# The Antitumor Agent Ansamitocin P-3 Binds to Cell Division Protein FtsZ in *Actinosynnema pretiosum*

**DOI:** 10.3390/biom10050699

**Published:** 2020-04-30

**Authors:** Xinran Wang, Rufan Wang, Qianjin Kang, Linquan Bai

**Affiliations:** 1State Key Laboratory of Microbial Metabolism, School of Life Sciences & Biotechnology, Shanghai Jiao Tong University, Shanghai 200240, China; shirley.wxr@sjtu.edu.cn (X.W.); wangrufan@sjtu.edu.cn (R.W.); qjkang@sjtu.edu.cn (Q.K.); 2Joint International Research Laboratory of Metabolic & Developmental Sciences, Shanghai Jiao Tong University, Shanghai 200240, China

**Keywords:** ansamitocin P-3, FtsZ, tubulin, toxicity, resistance

## Abstract

Ansamitocin P-3 (AP-3) is an important antitumor agent. The antitumor activity of AP-3 is a result of its affinity towards β-tubulin in eukaryotic cells. In this study, in order to improve AP-3 production, the reason for severe growth inhibition of the AP-3 producing strain *Actinosynnema pretiosum* WXR-24 under high concentrations of exogenous AP-3 was investigated. The cell division protein FtsZ, which is the analogue of β-tubulin in bacteria, was discovered to be the AP-3 target through structural comparison followed by a SPR biosensor assay. AP-3 was trapped into a less hydrophilic groove near the GTPase pocket on FtsZ by hydrogen bounding and hydrophobic interactions, as revealed by docking analysis. After overexpression of the *APASM_5716* gene coding for FtsZ in WXR-30, the resistance to AP-3 was significantly improved. Moreover, AP-3 yield was increased from 250.66 mg/L to 327.37 mg/L. After increasing the concentration of supplemented yeast extract, the final yield of AP-3 reached 371.16 mg/L. In summary, we demonstrate that the cell division protein FtsZ is newly identified as the bacterial target of AP-3, and improving resistance is an effective strategy to enhance AP-3 production.

## 1. Introduction

Ansamitocin P-3 (AP-3) is an important antitumor agent with potent activity against various cancer cell lines [1,2]. Since 2012, AP-3 has been used as the ‘warhead’ molecule of the currently commercialized antibody-conjugated (ADC) drug trastuzumab emtansine (T-DM1) for the treatment of breast cancer [3]. T-DM1 is also under clinical trials for the treatment of HER2-positive lung cancer [4].

In eukaryotic cells, binding of AP-3 on β-tubulin inhibits microtubule assembly and chromosome segregation during the interphase and mitotic phase resulting in cell apoptosis [5]. AP-3 has a strong interaction with β-tubulin with a dissociation constant (*K*_D_) of 1.3 ± 0.7 µM [5]. In a high-resolution crystal structure of tubulin in complex with maytansine, an AP-3 analogue [6], it was found that binding of maytansine interferes the interaction of longitudinal tubulin in microtubule assembly. The identified maytansine-binding site is distinct from the vinblastine-binding site, and is also shown as the common binding site of a rhizoxin variant and PM060184 [6].

Since tubulin is not present in bacterial cells, the intracellular targets of AP-3 in bacteria are still unknown. The cell division protein FtsZ has been shown to have similar function and architecture with tubulin [7,8,9]. Both FtsZ and tubulin work as cytoskeletons and form filaments with GTP hydrolysis [10,11]. Several compounds, such as curcumin, taxanes and plumbagin, are found to inhibit the function of tubulin as well as FtsZ [9]. Whether AP-3 is also an inhibitor of FtsZ remained to be studied.

As an important drug precursor, various strategies have been applied to improve AP-3 yield [12,13,14,15,16,17]. Previously in our lab, a high-yield AP-3 producing strain *Actinosynnema pretiosum* NXJ-24 was constructed by engineering the post-modification steps, which resulted in a 5-fold increase in AP-3 yield to 246 mg/L [14]. However, the AP-3 yield could not be further improved despite several attempts, suggesting a cell inhibition or toxicity effect caused by high concentrations of AP-3. 

In this study, the cell toxicity of AP-3 against its producing strain was observed and investigated. The interaction of AP-3 with FtsZ was validated by SPR biosensor analysis, in vitro assembly assay and docking analysis. Further overexpression of FtsZ improved the resistance of the resulting strain and the yield of AP-3 as well. 

## 2. Materials and Methods

### 2.1. Strains, Plasmids and Media

*A. pretiosum* strains were cultured on YMG plates (0.4% yeast extract, 1% malt extract, 0.4% glucose and 1.5% agar). For fermentation, strains were initially cultured on YMG plates at 30 °C for 48 h and then inoculated into S1 medium (0.5% yeast extract, 3% tryptone soya broth and 10.3% sucrose) to grow at 30 °C, 220 rpm for 24 h. S1 culture was subsequently transferred at 3.3% (*v*/*v*) into S2 medium (0.8% yeast extract, 3% tryptone soya broth, 10.3% sucrose, isobutanol 500 μL/L and isopropanol 500 μL/L, pH 7.5) and cultivated for another 24 h. S2 was transferred at 10% (*v*/*v*) into the fermentation medium (1.6% yeast extract, 1% malt extract, 10.3% sucrose, isopropanol 12 mL/L, isobutanol 5 mL/L, MgCl_2_ 2 mM and L-valine 40 mM, pH 7.5) and cultivated at 25 °C, 220 rpm for 10 days. The composition of the optimized fermentation medium is: 2.4% yeast extract, 1% malt extract, 10.3% sucrose, isopropanol 12 mL/L, isobutanol 5 mL/L, MgCl_2_ 2 mM and L-valine 40 mM, pH 7.5. For *E. coli* strains, Luria-Bertani (LB) broth was used. The information of strains, plasmids and primers used in this study are listed in Appendix A.

### 2.2. Growth Determination of Actinosynnema pretiosum Using Microplate Reader

Growth of *A. pretiosum* was determined using a microplate reader. During fermentation, 100 μL of culture was collected each day and mixed with 900 μL 10.3% sucrose. Then 50 μL of the mixture was transferred into a well of a 96-well plate. The fermentation medium without inoculation was used as a negative control. For each well, the absorbance was calculated by averaging the readings from 21 different positions of the well at 600 nm. 

### 2.3. AP-3 Resistance Evaluation 

Resistance evaluation on solid medium was performed with petri dishes (d = 3.5 cm). Before resistance evaluation, glycerol stocks of WXR-24 (high-yield AP-3 producer of *A. pretiosum)* and WXR-30 (*ftsZ*-overexpressing mutant of *A. pretiosum*) were diluted with 20% glycerol to have around 100 colony forming units (CFU) per 1 μL. For resistance evaluation, each Petri dish was filled with 4 mL YMG solid medium containing different concentrations of AP-3. Then 20 μL diluted stock of WXR-24 or WXR-30 were spread on the dishes. The dishes were incubated at 30 °C for 72 h and taken out for pictures and colony counting. 

To evaluate strain resistance in liquid medium, WXR-24 and WXR-30 were initially cultivated on YMG plates for two days, followed by consecutive cultivation in S1 and S2 media. Then 150 μL of S2 was transferred into a well of 24-deep-well plate, in which 1.5 mL fermentation medium containing different concentrations of AP-3 was added. For growth comparison of WXR-24 and WXR-30, the S2 cultures of the two strains were diluted to obtain similar CFUs before inoculation. Strain resistance was evaluated by growth determination from day 1 to day 5 using a microplate reader. 

For AP-3 supplementation, AP-3 was dissolved in DMSO, and the solution was added to the solid or liquid media before inoculation. Media supplied with same amount of DMSO was used as a control. The final concentration of DMSO was 1.5% (*v*/*v*). 

### 2.4. Colony Counting and Average Colony Size 

The numbers of colonies (CFU) were counted by the scientific image analysis software IMAGE J 1.52a (NIH, Bethesda, MD, USA). For each Petri dish, the 2-mm edge was removed to avoid light interruption. For average colony size analysis, particles with sizes from 5 to 500 pixel^2^ were considered to obtain an average colony size (pixel^2^) of each dish, with SD. The true colony size was calculated by converting the pixel^2^ into the actual size with a ratio of pixel-to-mm, obtained from the ‘scale’ function. 

### 2.5. Homologous Modeling and Structure Comparison 

The 3D structure of FtsZ in *A. pretiosum* NXJ-24 was built according to the structure of *Mycobacterium tuberculosis* (Protein Data Bank (PDB) entry 5v68) by homologous modeling with Discovery Studio 3.5 (Accelrys, San Diego, CA, USA). The structure of β-tubulin was obtained from Protein Data Bank (PDB) entry 4TV8, chain D. The structures of FtsZ and β-tubulin were compared using the FATCAT server [18]. A flexible alignment model was chosen for structure comparison. 

### 2.6. Protein Expression and Purification 

Gene *APASM_5716*, coding for the cell division protein FtsZ in *A. pretiosum* WXR-24, was amplified by PCR using primers 5716-28a-FP and 5716-28a-RP (Appendix A), sequenced and inserted into the *Nde*I/*Eco*RI sites of the vector pET28a. The resulted plasmid was introduced into *E. coli* BL21(DE3) for protein expression. The resulting strain was cultivated at 37 °C to OD_600nm_ of 0.6–0.8, then induced with 0.1 mM IPTG and cultivated at 30 °C for another 12 h. Cells were then harvested by centrifugation and re-suspended in 10 mM PBS (pH 7.5). After sonication, cell debris were removed by centrifugation at 11,000 rpm for 30 min on a Thermo Sorvall ST 16R centrifuge (ThermoFisher, Waltham, MA, USA). FtsZ was purified by nickel-NTA affinity chromatography after elution with PBS containing 50 mM–250 mM imidazole. The 250 mM imidazole eluent was ultra-filtrated into a volume of 500 μL, and then 10 mL corresponding buffer for the following assays was added, followed by ultra-filtration again into 500 μL. For SPR molecule interaction determination, the eluent after nickel-NTA purification was ultra-filtrated and directly injected into fast protein liquid chromatography (FPLC) equipped with a Superdex 200 gel filtration column (GE Healthcare Life Sciences, Marlborough, MA, USA) for further purification [19]. The purified proteins were used for analysis or stored at −80 °C for future use.

### 2.7. Surface Plasmon Resonance (SPR) Biosensor Analysis 

The interaction and dissociation constants between FtsZ and AP-3 were determined by surface plasmon resonance biosensor analysis using Biacore^TM^ 8K biosensor (GE Healthcare) at 25 °C. For molecule interaction determination, the purified FtsZ was bound to the Series S sensor chip CM5. In detail, 20 μg/mL FtsZ in 10 mM NaAc (pH 4.0) was coupled to the CM5 chip at 6000 RU, then AP-3 solutions of 0–100 μM were flew through the chip, and signal responses were recorded. The signal of solution without AP-3 was used as a blank control, and the value was subtracted from other values during data processing.

### 2.8. Light Scattering and GTPase Activity Assays

In vitro assembly of FtsZ was monitored on a F-7000 fluorescence spectrometer (Hitachi, Tokyo, Japan). Purified FtsZ was ultra-filtrated into J buffer (50 mM KCl, 50 mM MES, pH 6.5, 10 mM CaCl_2_, 5 mM MgCl_2_). 15 μM FtsZ was used for light scattering assay. The reaction was performed at room temperature and initiated upon the addition of 1 mM GTP. The emission and excitation wavelengths were set at 350 nm, and the slit width was 5 nm. Scatter signals were recorded every 2 min. 

GTPase activity was determined using Malachite Green Phosphate Assay Kit (BioAssay Systems, Hayward, CA, USA) following the manufacturer’s instructions. To test if the supplementation of AP-3 affected FtsZ assembly and GTPase activity, 500 μM AP-3 was used, and same amount of DMSO was added as control. 

### 2.9. Gene Overexpression

*APASM_5716* was amplified by PCR using primers 5716-646-FP and 5716-646-RP (Appendix A) and inserted into the *Nde*I/*Eco*RI sites of a pSET152-derived shuttle vector pLQ646 using Multi One Step Cloning Kit (Yeasen, Shanghai, China) (Appendix A) [20]. The resulted plasmid was sequenced and transformed to *E. coli* ET12567(pUZ8002) [19]. Then it was introduced into *A. pretiosum* WXR-24 by conjugation according to protocol described elsewhere [21]. 

### 2.10. AP-3 Yield Determination

For yield determination, two volumes of methanol was added to the final culture after fermentation. The mixture was sonicated for 30 min. Then 1 mL of the mixture was collected and centrifuged, and the supernatant was filtered through a 0.22-μm filter and subjected to HPLC analysis. The yields of AP-3 were analyzed according to the standard curve (Appendix A).

For HPLC determination, samples were analyzed on an Agilent series 1260 HPLC system (Agilent Technologies, Santa Clara, CA, USA) equipped with an Agilent Eclipse Plus C_18_ column (4.6 × 250 mm, 5 µm), eluted with 45% A (H_2_O with 0.5% formic acid):55% B (acetonitrile) for 10 min and detected at 254 nm. 

### 2.11. Docking Analysis

Molecule docking was performed using Discovery Studio 3.5 (Accelrys). AP-3 structure was obtained by modification of a maytansine structure from the co-structure of tubulin and maytansine from PDB entry 4TV8. Before docking, the ligands were prepared using the “prepare ligand” module. The ligands were primarily positioned in the binding site using LibDock, which is a high-throughput docking algorithm that positions catalyst-generated ligand conformations in the protein active site based on polar interaction sites (hotspots) [22]. Other algorithms such as GOLD and CDOCKER generated random conformations using CHARMm-based molecular dynamics. FtsZ was enclosed in a sphere grid with a radius of 15 Å. The Lamarckian genetic algorithm was used. Total of 15 runs were performed, and 32 output conformations were obtained. A cutoff RMSD of 2.14 Å was employed to cluster all conformations, and clusters having lowest docking energy were considered. The residues in the interphase of two FtsZ monomers were identified by sequence alignment with the model of *Methanocaldococcus jannaschii* (PDB number 1w5a).

### 2.12. Microscopy Analysis

The morphology of mycelia was observed using a Revolve-100-G microscope (ECHO, San Diego, CA, USA). For sampling, 1 mL culture of WXR-24 or WXR-30 was collected, washed twice and resuspended in 10 mM PBS (pH 7.5). Ten μL of the mycelial suspension was mixed with 50 μL H_2_O on a standard glass slide (25 × 75 mm). After immobilization, the sample was dyed with crystal violet staining solution. The images were captured using digital camera with 20- or 60-fold magnification under the optical microscope mode. 

### 2.13. qRT-PCR Analysis

RNA was extracted with the Redzol reagent (SBS Genetech, Beijing, China) and converted to cDNA using the RevertAid^TM^H Minus First Strand cDNA Synthesis Kit (ThermoFisher), according to the manufacture’s instruction. RT-PCR was performed on a QuantStudio^TM^ 3 Real-Time PCR Instrument (Applied Biosystems, Carlsbad, CA, USA) using TB Green *Premix Ex Taq*^TM^ GC enzyme (Takara, Shiga, Japan). The gene transcription was calculated using 2^-^^△△^^CT^ method, and the sigma factor gene *hrdB* was set as the internal control. 

## 3. Results

### 3.1. Growth Inhibition of Actinosynnema pretiosum WXR-24 Under Exogenous AP-3 Supplementation 

The previously constructed high-yield strain NXJ-24 has an integrated copy of the pSET152-based plasmid (pLQ586) in the genome [14]. For stable fermentation and convenience of further engineering, an alternative strain WXR-24 was constructed by introducing the overexpressing cassette (*kasOp*-asm10*) from pLQ586 to the chromosome by double crossover recombination (Appendix A). To test whether WXR-24 has the potential to be a high-yield AP-3 producer, its tolerance to high concentrations of AP-3 was evaluated. The growth of WXR-24 was determined on YMG plates containing 0, 100, 200, 300 and 400 mg/L AP-3. After 3 days, no decrease in survival rate was observed under supplementation of 100 mg/L AP-3. However, when the AP-3 concentration increased to 200 mg/L, the survival rate decreased to 72%. No growth occurred on plates with 300 mg/L and 400 mg/L AP-3 (Figure 1a,b and Appendix A). After 16 days of incubation, slow growth was observed on plates containing 300 mg/L and 400 mg/L AP-3, with 150 and 11 CFUs, respectively (Appendix A). In addition, the average colony size of WXR-24 decreased 38.4% with supplementation of 100 mg/L AP-3, and a further decrease of 66.0% was observed with 200 mg/L AP-3 (Appendix A), suggesting growth inhibition of WXR-24 by AP-3 on YMG plates.

Cell growth in liquid medium was also studied. In the medium without AP-3 supplementation, OD_600_ of WXR-24 increased rapidly from 0.05 to 0.25 during cultivation. While the initial OD_600_ was unchanged, the final OD_600_ decreased from 0.25 to 0.22, 0.20, 0.11 and 0.08 in media supplemented with 100, 200, 300 and 400 mg/L AP-3, respectively (Figure 1c,d). These results indicate that the growth of WXR-24 is also inhibited in liquid culture by AP-3 supplementation. 

### 3.2. Cell Division Protein FtsZ is Predicted as a Target of AP-3

Since the growth of WXR-24 was inhibited by AP-3 supplementation, in order to understand the mechanism, it is necessary to identify the intracellular targets of AP-3. In eukaryotic cells, AP-3 binds to β-tubulin and consequently inhibits microtubule assembly and cell division [7]. Because tubulins are not present in *A. pretiosum*, it was therefore inferred that FtsZ, as a structural homolog to tubulin [23], is likely to be the intracellular target of AP-3. To confirm the hypothesis above, the two structures were compared.

The structure of FtsZ in WXR-24 was simulated by homologous modeling based on the structure of FtsZ from *Mycobacterium tuberculosis* (Protein Data Bank (PDB) entry 5v68). The structure of β-tubulin was obtained from the maytansine-tubulin complex (PDB entry 4TV8, chain D). Both proteins consist of twelve α-helices spaced by ten β-sheets. The N-terminal and C-terminal domains of the two proteins are divided by the featured α-helix 7 located horizontally in the middle of the proteins (Figure 2a,b). 

These two structures were further compared according to the flexible structure alignment algorithm [18]. The result showed that they are significantly similar with a *p*-value of 2.56 × 10^−7^ (raw score is 483.77). The two structures have 333 equivalent positions with an RMSD of 3.02, and 2 positions could not be completely aligned. Three large conserved blocks made up of the 333 equivalent positions are formed (colored in red, yellow or pink), which are separated by the two non-homologous positions (Figure 2c). In detail, structures from S1 to S10 (red, corresponding to amino acid residues 31–331 of FtsZ and 1–381 of β-tubulin), H11 (yellow, residues 347–357 of FtsZ and 382–400 of β-tubulin) and H12 (pink, residues 401–420 of FtsZ and 415–438 of β-tubulin) are highly similar between the two proteins, but structure between S10 and H11 and that between H11 and H12 could not be aligned (Figure 2d,e). In the superposition of FtsZ and β-tubulin, GDP pockets are located in similar positions of the two proteins (Figure 2e). Although most key amino acid residues in β-tubulin interacting with maytansine are absent in FtsZ (Figure 2d), similar structure of the maytansine-binding pocket still exists in FtsZ (Figure 2e). Based on these comparison, FtsZ was considered to be the intracellular target of AP-3. 

### 3.3. In Vitro Interaction of FtsZ and AP-3

To test if AP-3 could bind to FtsZ, FtsZ protein from *A. pretiosum* WXR-24 was heterologously expressed in *E. coli* and purified. The interaction of FtsZ with AP-3 was determined by surface plasmon resonance (SPR) biosensor analysis. Before analysis, 6000 RU of FtsZ was immobilized on a CM5 chip, then AP-3 solutions with concentrations of 6.25, 12.5, 25, 50 and 100 μM successively flew over the FtsZ conjugated surface. The injection of 6.25–100 μM AP-3 induced a gradually increased response of the FtsZ-bound surface, indicating a stable binding between FtsZ and AP-3 (Figure 3a). The binding ratio of FtsZ and AP-3 was 1:1, and the association rate and dissociation rate constants *K*a and *Kd* were 2.36 × 10^3^ M^−1^s^−1^ and 0.794 s^-1^, respectively. The equilibrium dissociation constant *K*_D_ was 3.36 × 10^-4^ M (Appendix A). Since the *K*_D_ of AP-3 and β-tubulin is 1.3 ± 0.7 × 10^-6^ M [5], the binding of AP-3 to FtsZ was thought to be much weaker than the binding of AP-3 to β-tubulin.

To study whether the supplementation of AP-3 affects FtsZ assembly, light scattering experiment was performed to monitor the real-time FtsZ assembly process. The supplementation of AP-3 had an obvious negative impact on in vitro FtsZ assembly (Figure 3b). GTPase activity of FtsZ was also studied under AP-3 supplementation. The GTPase activity of FtsZ was slightly decreased upon AP-3 addition (Figure 3c), suggesting that FtsZ assembly is inhibited by AP-3 binding in a GTPase-independent manner. 

### 3.4. Docking Analysis of AP-3 on FtsZ 

Since AP-3 was confirmed to bind to FtsZ and inhibit FtsZ assembly, it is necessary to explore the interaction between AP-3 and FtsZ at the molecular level. For this purpose, docking analysis was performed around the putative maytansine-binding structure on FtsZ obtained from the structure comparison in Section 3.2. Fifteen independent runs were performed, generating 32 output conformations. Simulated conformation with the lowest energy of -19.47 kcal/mol was selected. 

AP-3 binding site is located near the N-terminal of α-helix 3, α-helix 4 and α-helix 5, and not far from the GDP pocket (Figure 4a). As shown in Figure 4b, the AP-3 binding site is at the edge of the junction region between two FtsZ monomers, which explains the inhibition of FtsZ assembly by AP-3 supplementation and the unaffected GTPase activity of FtsZ. Whereas the surface of FtsZ is highly hydrophilic, the AP-3 binding pocket is comprised of less hydrophilic amino acids (Figure 4c). The macrolactam ring of AP-3 is inserted into the hydrophobic pocket, and key amino acid residues of FtsZ interacting with AP-3 are identified as Pro92, His96, Gly129, Gln169 and Asn173. In detail, the carbinolamide ring of AP-3 is attached to the amine groups of Gln169 and Asn173 with hydrogen bonds, and the nearby oxygen atom on C-10 of AP-3 forms hydrogen bond with the amine group on His96. The other side of the macrolactam ring attaches to the pocket via hydrophobic interactions between the carbon skeleton of C14-C15 and Pro92 and Gly129 (Figure 4d). 

### 3.5. Introducing an Extra Copy of ftsZ Improved AP-3 Resistance and Production

Since FtsZ was identified as the target of AP-3, we hypothesized that its overexpression might be able to alleviate the toxicity of AP-3 to the growth of WXR-24. To test this, *APASM_5716*, coding for FtsZ in *A. pretiosum*, was overexpressed under the strong promoter *kasOp** on the integrative plasmid pLQ646 (Appendix A). The resulting plasmid pLQ1913 was introduced into WXR-24 by conjugation, and the resulting strain was named as WXR-30. The transcriptional level of *ftsZ* had a 202% increase in WXR-30 (Appendix A). As shown in Appendix A, WXR-30 exhibited superior growth than WXR-24 during cultivation. Microscopic observation showed that, with the overexpression of *ftsZ*, the mycelial morphology of *A. pretiosum* was changed, the branches became shorter and the long hyphae intertwined with each other to form clumps (Appendix A).

The resistance of WXR-30 was evaluated after 3 days of incubation, and the cell survive rate was 100% under supplementation of 100 mg/L AP-3. (Figure 5a,b and Appendix A). With supplemented 200 mg/L AP-3, the survive rate of WXR-30 showed a slight decrease to 95%, which was obviously higher than that of WXR-24 under the same conditions (72%). Moreover, 14 and 8 colonies appeared on plates supplemented with 300 mg/L and 400 mg/L AP-3, respectively. After 16-day incubation, 1080 and 1302 colonies appeared on plates supplemented with 300 mg/L and 400 mg/L AP-3, respectively (Appendix A), which was 7-fold and 118-fold those of WXR-24 under the same conditions. In addition, the average colony sizes of WXR-30 decreased by 31% and 38% when supplemented with 100 and 200 mg/L of AP-3, respectively (Appendix A), substantially lower than those of WXR-24 (38% and 66%), suggesting a better growth of WXR-30 under AP-3 supplementation conditions. 

Cell growth of WXR-30 in liquid medium was also studied. The initial OD_600_ of WXR-24 and WXR-30 were both 0.05. Whereas the final OD_600_ of WXR-24 was 0.11, the final OD_600_ of WXR-30 reached 0.21, showing a significantly improved growth of WXR-30 in liquid culture (Figure 5c).

The effect of FtsZ overexpression on AP-3 production was also evaluated with WXR-30. Compared with the starting strain WXR-24, the yield of AP-3 for WXR-30 was increased by 31%, from 250.66 mg/L to 327.37 mg/L (Figure 5d). In addition, the concentrations of supplemented yeast extract were optimized with WXR-30. When 24 g/L yeast extract was used in the fermentation medium, the AP-3 yield of WXR-30 was further improved to 371.16 mg/L (Figure 5d and Appendix A), i.e., 48% higher than that of WXR-24. 

## 4. Discussion

In this study, FtsZ was identified as the intracellular target of AP-3 in *A. pretiosum.* It was initially believed that AP-3 is not toxic to bacteria, because tubulin does not exist in bacteria. However, since the bacterial cell division protein FtsZ was reported to have a similar structure to tubulin, it was inferred that FtsZ might be a target of AP-3 [7,8,9]. In this study, structures of FtsZ from *A. pretiosum* and β-tubulin from *Bos taurus* were compared and shown to be structurally similar to each other [18]. In a subsequent in vitro interaction test, it was validated that AP-3 could bind to FtsZ and inhibit FtsZ assembly. This is the first time that the bacterial target of AP-3 has been identified. Our further experiments found that AP-3 could also inhibit the assembly of FtsZ from *Bacillus subtilis* and *Streptomyces coelicolor* (Appendix A), indicating that the inhibition of AP-3 on FtsZ is not limited in *Actinosynnema* but broadly applicable among bacteria. 

Some antitumor antibiotics also showed cell toxicities towards their bacterial producers [24,25,26]. During the production of antitumor tiancimycin A, for example, its accumulation was extremely toxic to its producer *Streptomyces* sp. CB03234, and the yield was only 0.2 mg/L [27]. In this study, it was observed that exogenous AP-3 at a concentration higher than 300 mg/L caused severe inhibition on the growth of WXR-24 (Figure 1c and Appendix A). Improving strain resistance has been widely used to enhance the yields of various antibiotics, such as oxytetracycline [28], doxorubicin [29], rimocidin [30], daptomycin [31] and toyocamycin [32]. However, these strategies mainly focus on overexpressing efflux genes or engineering resistance genes. As another option, overexpressing the drug targets has also been long used to alleviate cell toxicities and improve resistance [33]. For example, the overexpression of *embAB* genes, coding for the targets of the antimycobacterial compound ethambutol (Emb), resulted in Emb resistance [34]. The overexpression of FDP synthase, target of aminobisphosphonate (aBP) drugs, improved resistance of aBP drugs in *Dictyostelium amoebas* [35]. In our study, overexpression of the AP-3 target FtsZ improved both strain resistance and AP-3 production, showing again that the overexpression of target proteins is an effective way to enhance antibiotic production. 

FtsZ plays a fundamental role in cell division of bacteria. Before cell division, a ring-like structure is formed in the middle of the dividing cell by FtsZ polymerization, which then recruits other related proteins to start the dividing program [9]. Increased expression of *ftsZ* often results in change of cell morphology. For example, in *B. subtilis* and *E. coli*, *ftsZ* overexpression will cause asymmetric cell division [36,37], while in *C. glutamicum* an increase in cell diameter is observed [38]. Improved expression of *ftsZ* in *S. coelicolor* and *S. lividans* inhibited aerial mycelia and spore formation, and antibiotic production was increased [39]. In our study, it was found the length of the branching hyphae in the *ftsZ*-overexpressing mutant WXR-30 was much shorter than the control strain (Appendix A). The result is in accordance with the work in *S. venezuelae*, in which highly branched mycelia were found in the *ftsZ* mutant [40]. The relationship between mycelial morphology and AP-3 resistance or production needs further investigation. 

The cell toxicity of AP-3 on its producer *A. pretiosum* was less pronounced than on eukaryotic cells. In fact, the IC_50_ of AP-3 on MCF-7 cells is 20 ± 3 pmol/L, which is much lower than that on *A. pretiosum* (315.0–472.4 μmol/L). The binding force of AP-3 with FtsZ (*K*_D_ = 3.36 × 10^−4^ M) (Appendix A) is also obviously weaker than that with β-tubulin (*K*_D_ = 1.3 × 10^−6^ M). As shown in Figure 4d, three hydrogen bonds are formed between AP-3 and Gly129, Asn173 and Gln169 on FtsZ, with bond lengths of 3.1, 5.9 and 3.9 Å, respectively. In comparison, five hydrogen bonds are formed between maytansine and Gly100, Asn102, Lys105, Asn101 and Val181 on β-tubulin, with bond lengths of 3.1, 3.2, 2.5, 3.3 and 2.9 Å, respectively. As hydrogen bonding is relatively strong among intermolecular interactions [41], the interaction between AP-3 and β-tubulin is therefore much stronger than that with FtsZ, which explains the toxicity of AP-3 towards *A. pretiosum* only at high concentrations.

Although the resistance of WXR-30 was substantially improved (Figure 4a), the strain was still sensitive to 300 mg/L and 400 mg/L AP-3, which implies that FtsZ overexpression could not fully confer resistance to AP-3. From our docking analysis, five key amino acid residues, namely Pro92, His96, Gly129, Gln169 and Asn173, were proposed to interact with AP-3 (Figure 4d). Therefore, these key residues, especially Gly129, Gln169 and Asn173 forming hydrogen bonds with AP-3, will be considered for future mutagenesis to further alleviate or even abolish the interactions and consequent cell toxicity. 

## 5. Conclusions

AP-3 toxicity was found to be a rate-limiting factor for high AP-3 production in *A. pretiosum* WXR-24. For the first time, FtsZ was identified as the target of AP-3 in bacteria. Over-expression of the FtsZ coding gene resulted in AP-3 resistance and overproduction. Our study suggests that toxicity is one of crucial factors limiting antibiotic production, especially in high-yield industrial strains, and resistance improvement is an effective way to further enhance antibiotic yields. 

## Figures and Tables

**Figure 1 biomolecules-10-00699-f001:**
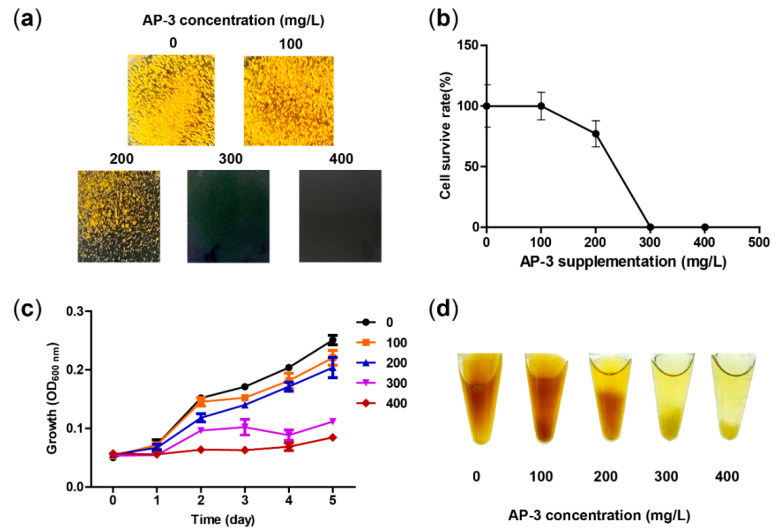
Resistance and growth of *A. pretiosum* WXR-24 under different concentrations of AP-3. (**a**) Growth of *A. pretiosum* WXR-24 on YMG plates containing 0, 100, 200, 300 or 400 mg/L AP-3. The photos were taken on day 3 after inoculation. (**b**) Survive rate (%) of WXR-24 on YMG plates containing 0, 100, 200, 300 or 400 mg/L AP-3. The survive rate of the plates supplemented without AP-3 was set as 100%, and the values on other plates were accordingly calculated. Mean values of three independent experiments are shown with SD indicated by error bars. (**c**) Growth of *A. pretiosum* WXR-24 in fermentation broths containing 0, 100, 200, 300 or 400 mg/L AP-3. Mean values of three independent experiments are shown with SD indicated by error bars. (**d**) The cultures of *A. pretiosum* WXR-24 on day 5 after inoculation. Less mycelia are observed in fermentation broths containing 200–400 mg/L AP-3.

**Figure 2 biomolecules-10-00699-f002:**
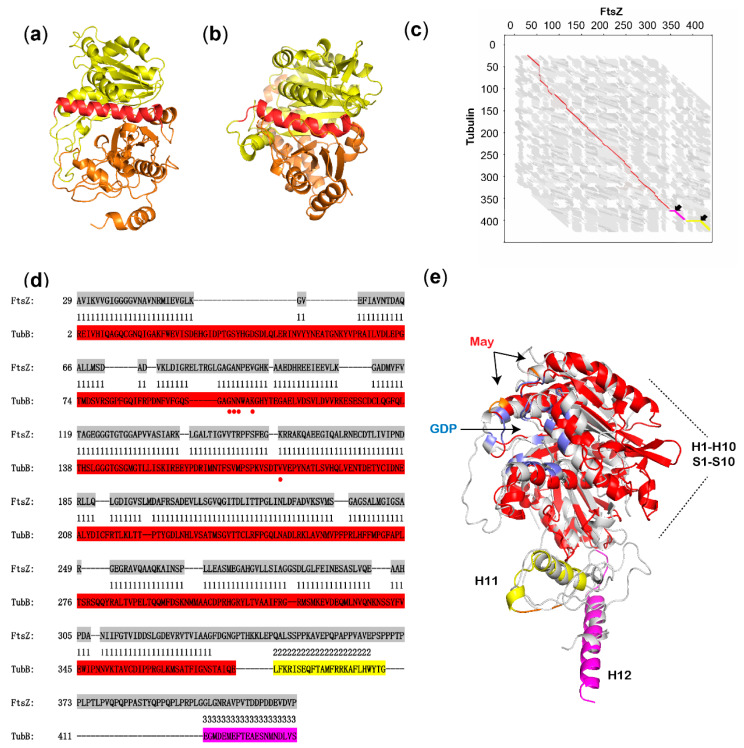
Structure comparison of FtsZ and β-tubulin. (**a**) The architecture of FtsZ from *A. pretiosum* WXR-24, built by homologous modeling. The featured α-helix 7 is labeled in red, connecting the yellow-labeled N-terminal domain and orange-labeled C-terminal domain. (**b**) The architecture of β-tubulin from *Bos taurus* retrieved from PDB database. The featured α-helix 7 is labeled in red, connecting the yellow-labeled N-terminal domain and orange-labeled C-terminal domain. (**c**) The graph of flexible structure alignment of FtsZ and β-tubulin. Three large structurally conserved blocks are shown in red, yellow or pink, and 2 twists are indicated by black arrows. (**d**) Sequence-based structural alignment of FtsZ and β-tubulin. FtsZ is highlighted in grey, and structurally conserved blocks in β-tubulin are highlighted in red, yellow or pink. The key amino acids of β-tubulin interacting with maytansine are indicated by red dots. (**e**) Superposition of the 3D structures of FtsZ and β-tubulin. FtsZ is colored in grey, and structurally conserved blocks in β-tubulin are in red, yellow or pink. The GDP-binding pockets of FtsZ and β-tubulin are labeled in blue, and the maytansine-binding pocket on β-tubulin is labeled in orange. May, maytansine.

**Figure 3 biomolecules-10-00699-f003:**
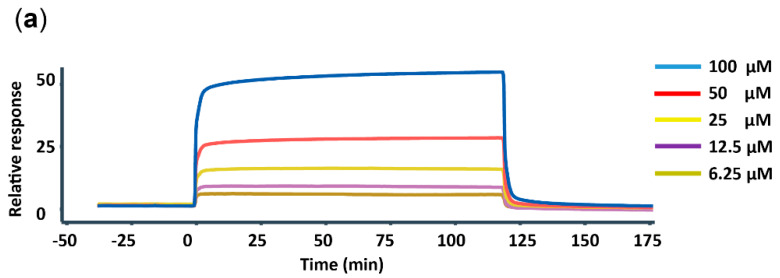
The in vitro interaction of FtsZ with AP-3. (**a**) The interaction of FtsZ with AP-3, determined by Biacore biosensor assay. 6.25, 12.5, 25, 50 or 100 μM AP-3 flowed over a CM5 chip pre-immobilized with FtsZ. The response signals were recorded. (**b**) The inhibition of FtsZ assembly by the supplementation of AP-3, as detected by the real-time light scattering assay. Fluorescence signals of the reactions with (red line) or without (black line) 500 μM AP-3 were recorded every 2 min. (**c**) The impact of AP-3 (500 μM) on the GTPase activity of FtsZ, as detected by the Malachite Green Phosphate Assay Kit. Black line, without AP-3; Red line, with AP-3.

**Figure 4 biomolecules-10-00699-f004:**
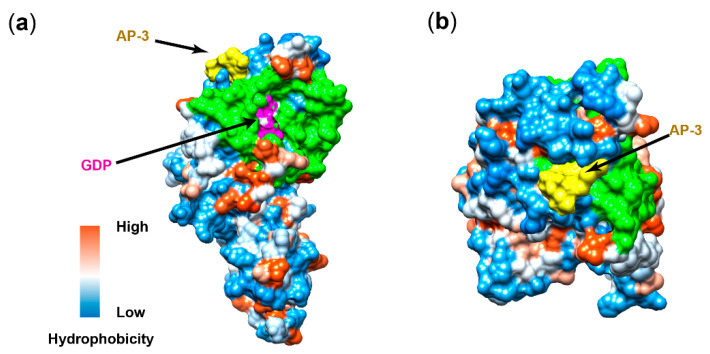
Interaction of FtsZ and AP-3 revealed by docking analysis. (**a**) The binding site of AP-3 on FtsZ is apart from the GDP binding site, from side view. The surface of FtsZ is displayed by hydrophobicity. Orange and blue indicate high and low hydrophobicity, respectively. The junction interphase between two FtsZ monomers is colored in green. (**b**) The binding site of AP-3 on FtsZ, from top view. (**c**) Close-up view of the AP-3 binding groove on FtsZ. The colors on the groove indicate that this part is more hydrophobic than the surrounding area. (**d**) The key amino acid residues interacting with AP-3. The interactions between key amino acids and AP-3 are displayed by dashed lines. Red dashed lines represent hydrogen bonds, and grey dashed lines represent hydrophobic interactions. The bond lengths (Å) are indicated by the numbers beside the dotted lines.

**Figure 5 biomolecules-10-00699-f005:**
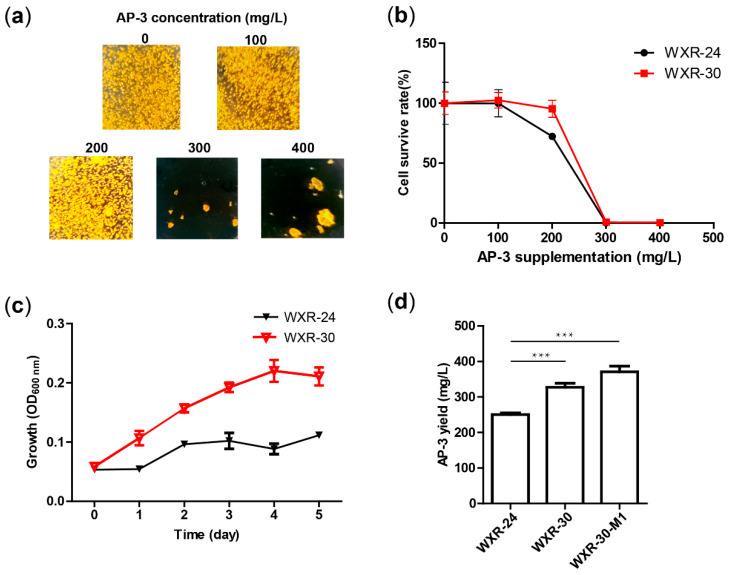
Effects of FtsZ overexpression on the resistance, growth and AP-3 yield of *A. pretiosum* WXR-30. (**a**) Growth of WXR-30 on YMG plates containing 0, 100, 200, 300 or 400 mg/L AP-3. The photos were taken on day 3 after inoculation. (**b**) Survive rate (%) of WXR-24 and WXR-30 on YMG plates containing 0, 100, 200, 300 or 400 mg/L AP-3. The survive rate on the plates supplemented without AP-3 was set as 100%, and the survive rates on other plates were accordingly calculated. Mean values of three independent experiments are shown with SD indicated by error bars. (**c**) Growth of *A. pretiosum* WXR-24 and WXR-30 in the fermentation broth containing 300 mg/L AP-3. Mean values of three independent experiments with SD are indicated by error bars. (**d**) AP-3 yields of WXR-24 and WXR-30 cultivated with 16 g/L yeast extract, and WXR-30 cultivated with 24 g/L yeast extract. Mean values of three independent experiments with SD are indicated by error bars. *** *p* < 0.001.

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
