# Peer review of "The Antitumor Agent Ansamitocin P-3 Binds to Cell Division Protein FtsZ in Actinosynnema pretiosum"

_biomolecules, 2020, doi:10.3390/biom10050699_

Round 1

Reviewer 1 Report

The article “The antitumor ansamitocin P-3 binds to cell division protein FtsZ in Actinosynnema pretiosum” by Wang et al. describes the identification of FtsZ as a bacteria target of AP-3. It is well written, but I’d recommend to do some changes in order to make the manuscript ready for publication:

  1. The main aim of the paper is to improve the yield of AP-3 production, which is briefly mentioned in the introduction, but I believe that it should be stated also in the abstract.
  2. Where is Ac. pretiosum living? Does it use the production of AP-3 as a fitness advantage towards other bacteria/archea?
  3. How many replicate plates were done for the resistance evaluation experiments? And how many tubes did you use for the experiment in liquid? The data presented are often given as 1 number, for example for the control conditions you measured 2403 colonies, but is this an average? Was the experiment done more than once? What is the SD? Perhaps we should have these raw number in the supplementary data (improving therefore table S2) and present data (coming from biological replicates) as ratio of treated vs control, maybe on a log scale (and displaying SD). With respect to the liquid experiment, how could you make sure that the initial number of cells were the same for all the different conditions employed?
  4. The same written above applies to the data on the strain WXR-30.
  5. What was used for the HPLC standard curve?
  6. The binding of AP-3 to FtsZ is 100 times weaker than tubulin. All the first experiments are done in mg/l, but what is the concentration in mM or µM? Is the concentration employed within the range of inhibition measured with Kd?
  7. What’s the toxicity of AP-3 on other archea/bacteria? Is it a lot lower than the one of Ac. Pretiosum?
  8. If you’d keep improving the strains. What concentration would you wish to reach? What would be feasible?

Reviewer 2 Report

Dear Editor,

The manuscript “The antitumor ansamitocin P-3 binds to cell division protein FtsZ in Actinosynnema pretiosum” describes a study in which authors found the target (FtsZ) for the antitumor ansamitocin P-3 and constructed a mutant overexpressing the target to increase ansamitocin P-3 production in the producer strain.

The story is clear, the experiments were logically performed, the results interesting for readers working on this kind of molecules or on optimization of production of secondary metabolites but in my opinion it needs some severe improvements.

Introduction

First, the assumption “Recently, the cell division protein FtsZ has been shown to have similar function and architecture with tubulin [7].” is unfair since older reports had described this homology. It is well known that FtsZ is the bacterial homolog of the eukaryotic tubulin. The authors, more honestly, should cite old references, for example

  1. FtsZ, a prokaryotic homolog of tubulin? Erickson HP. Cell. 1995 Feb 10;80(3):367-70. doi: 10.1016/0092-8674(95)90486-7
  2. Amos LA, van den Ent F, Löwe J Curr Opin Cell Biol. 2004 Feb;16(1):24-31. Structural/functional homology between the bacterial and eukaryotic cytoskeleton. DOI: 10.1016/j.ceb.2003.11.005.

Thus, they should improve the introduction adding these references.

Secondly, since that it was very likely that FtsZ was the target of ansamitocin P-3, in my opinion the main result of the manuscript is the improvement of the resistance as an effective strategy to enhance AP-3 production. Thus, nothing is reported on resistance to this molecule in the introduction.

Material and Methods

Actinosynnema pretiosum should be reported as A. pretiosum not Ac. pretiosum along the whole manuscript and the authors should pay attention to the capitals.

For experiments reported in 2.2. Growth determination of Actinosynnema pretiosum using microplate reader, I suppose that negative and positive controls were added in the experiment. Could they add the controls?

In line 74, WXR-24 and WXR-30 are named without a brief explanation; the authors could write they are two mutants of A. pretiosum.

In line 75, CFU is mentioned for the first time, but it is explained in line 89.

For PCR experiments (2.6 and 2.8 paragraphs), the authors should indicate the name of the primers used and that the sequence is in table S1, where more details should be given to explain the underlined nucleotides.

The centrifuge should be indicated when rpm is used, otherwise xg is more appropriate.

Results

No experiments with the control strain are reported. But, more importantly, what in my opinion is missing, the amount of ansamitocin P-3 produced by the strains (both wt and mutants) and the phenotype analysis of the FtsZ overexpressing strain. Usually, FtsZ overexpression has an impact on cell morphology. How many copies of FtsZ should be produced in the FtsZ overexpressing strain, was a qRT-PCR analysis performed?

A better explanation on the construction of WXR-24 should be given, for example, asm10 gene is neither described or a reference is given.

Line 176, in is repeated twice

Regarding figure 1 and 5, I would remove the pictures of the plates.

Paragraph 3.2. Cell division protein FtsZ is predicted as a target of AP-3 should be rewritten in the light of the consideration that FtsZ is the homolog of tubulin.

Figure 2 (c) and (d) are too small and not well resolved.

For the Biacore biosensor assay, I suppose that a control with empty solutions or another protein was carried out. Can the authors add it?

The difference in the real-time light scattering assay, is enough to indicate the inhibition of FtsZ assembly? Which control was used?

Line 281, 282, I think the sentence “The light blue and orange indicate the groove is less hydrophilic than the surrounding surface of FtsZ” is not properly written.

In fig. 5d the production of the metabolite from the control strain is missing.

Discussion

Please rewrite the initial part “It was initially believed that AP-3 is not toxic to bacteria, because tubulin does not exist in bacteria. Structure analogue to microtubule was then discovered in bacterial cells. The cell division protein FtsZ in the hyperthermophilic methanogen Methanocaldococcus jannaschii, which also forms cytoskeletons inside the cell, was found to have similar structure with tubulin [7].”

The authors should comment on the importance and role of FtsZ in bacteria and the impact of its overexpression.

Table S1

The authors should pay more attention to the capitals. 

In table S2 cell survive rates is percentage? It should be indicated.

IN addition how can the authors exclude mutations after 16 days?

Round 2

Reviewer 1 Report

The authors have answer clearly to all my questions and have improved the manuscript accordingly.

Reviewer 2 Report

The authors addressed all my issues and I think the manuscript was improved.

Thus, in my opinion it deserves to be published.

best regards